# Molecular Assessment of HER2 to Identify Signatures Associated with Therapy Response in HER2-Positive Breast Cancer

**DOI:** 10.3390/cancers14112795

**Published:** 2022-06-04

**Authors:** Adam L. Maddox, Matthew S. Brehove, Kiarash R. Eliato, Andras Saftics, Eugenia Romano, Michael F. Press, Joanne Mortimer, Veronica Jones, Daniel Schmolze, Victoria L. Seewaldt, Tijana Jovanovic-Talisman

**Affiliations:** 1Department of Molecular Medicine, Beckman Research Institute, City of Hope Comprehensive Cancer Center, Duarte, CA 91010, USA; amaddox@coh.org (A.L.M.); mbrehove@chromologic.com (M.S.B.); kreliato@coh.org (K.R.E.); asaftics@coh.org (A.S.); eromano@coh.org (E.R.); 2Department of Pathology, Keck School of Medicine of the University of Southern California, Los Angeles, CA 90089, USA; michael.press@med.usc.edu; 3Department of Medical Oncology, City of Hope Comprehensive Cancer Center, Duarte, CA 91010, USA; jmortimer@coh.org; 4Department of Surgery, City of Hope Comprehensive Cancer Center, Duarte, CA 91010, USA; vjones@coh.org; 5Department of Pathology, City of Hope Comprehensive Cancer Center, Duarte, CA 91010, USA; dschmolze@coh.org; 6Department of Population Sciences, Beckman Research Institute, City of Hope Comprehensive Cancer Center, Duarte, CA 91010, USA; vseewaldt@coh.org

**Keywords:** super-resolution microscopy, breast cancer, trastuzumab, therapy response, HER2, clustering

## Abstract

**Simple Summary:**

The HER2 status of breast cancers is accurately determined by measuring HER2 protein overexpression and gene amplification. However, these clinical diagnostic tests cannot predict the response to therapy. Single molecule imaging approaches can quantify molecular features of HER2, such as receptor nano-organization, with exquisite spatial resolution and sensitivity. The aim of our study was to assess how the molecular features of HER2 varied with the therapy response. According to our results in cultured cell lines and six patient specimens, the therapy response was associated with high detected HER2 densities and clustering. This advanced imaging approach can thus provide key data to complement the current diagnostic standards.

**Abstract:**

Trastuzumab, the prototype HER2-directed therapy, has markedly improved survival for women with HER2-positive breast cancers. However, only 40–60% of women with HER2-positive breast cancers achieve a complete pathological response to chemotherapy combined with HER2-directed therapy. The current diagnostic assays have poor positive-predictive accuracy in identifying therapy-responsive breast cancers. Here, we deployed quantitative single molecule localization microscopy to assess the molecular features of HER2 in a therapy-responsive setting. Using fluorescently labeled trastuzumab as a probe, we first compared the molecular features of HER2 in trastuzumab-sensitive (BT-474 and SK-BR-3) and trastuzumab-resistant (BT-474^R^ and JIMT-1) cultured cell lines. Trastuzumab-sensitive cells had significantly higher detected HER2 densities and clustering. We then evaluated HER2 in pre-treatment core biopsies from women with breast cancer undergoing neoadjuvant therapy. A complete pathological response was associated with a high detected HER2 density and significant HER2 clustering. These results established the nano-organization of HER2 as a potential signature of therapy-responsive disease.

## 1. Background

The survival of women with HER2-positive breast cancer has been dramatically improved with the advancement of trastuzumab therapy [1,2,3]. Today, the first-line therapy regimen includes chemotherapy combined with HER2-directed therapy, monoclonal antibodies (mAbs) trastuzumab and pertuzumab. Eligibility for HER2-directed therapy is currently evaluated using a set of guidelines, which are determined by the American Society of Clinical Oncologists (ASCO) and the College of American Pathologists (CAP). Two assays are employed to assess the levels of HER2 in breast cancers: immunohistochemistry (IHC) and fluorescence in situ hybridization (FISH) [4,5,6,7,8]. IHC and FISH are highly accurate for predicting breast cancers that will not respond to HER2-directed therapies (negative-predictive marker) [9]. However, IHC and FISH have poor positive-predictive accuracy. Only 40–60% of women with HER2-positive breast cancers will achieve a complete pathological response to standard regimens of chemotherapy combined with either trastuzumab alone or with trastuzumab plus pertuzumab [10,11,12,13,14,15,16,17,18,19]. During 10 years of clinical follow-ups, approximately 20% of women will either experience disease recurrence or succumb to disease [20,21]. A broad range of new therapies are being developed to target the HER-family of receptors. Since many options now exist, new assays are needed, not only to identify non-responders (negative predictor), but also to identify who will benefit with high positive-predictive accuracy.

Trastuzumab was the first mAb approved by the U.S. Food and Drug Administration (FDA) to treat HER2-postive breast cancers [22,23]. The efficacy of this first-line neoadjuvant therapy has been linked to four antitumor effects [24,25,26,27,28,29]: (1) the rapid internalization and degradation of HER2, (2) the disruption of HER2 homodimerization, (3) the inhibition of extracellular HER2 proteolytic cleavage, and (4) the stimulation of antibody-dependent cell-mediated cytotoxicity (ADCC). The efficacy of trastuzumab has led to several trastuzumab-based drug conjugates. Two FDA-approved conjugates, trastuzumab emtansine (T-DM_1_) and trastuzumab deruxtecan (T-DXd), have shown improved antitumor properties and overall survival [17,30,31,32]. While trastuzumab-based therapies work by binding to domain IV of HER2 (Appendix A), another FDA-approved anti-HER2 mAb, pertuzumab, targets domain II of HER2 (Appendix A). It not only inhibits HER2 heterodimerization but also blocks important mitogenic signaling cascades [33,34]. Altogether, these mAb-based HER2-directed therapies have significantly improved patient outcomes [1,2,33,35,36,37,38,39,40,41,42,43]. 

According to a recent phase 3 study (1196 patients from 25 countries), the 8-year landmark overall survival rate for patients on chemotherapy combined with trastuzumab and pertuzumab was 37%, and for patients on chemotherapy combined with trastuzumab alone, it was 23% [44]. Because first-line treatment of HER2-positive disease includes chemotherapy combined with HER2-directed agents, one cannot distinguish whether the cancer is sensitive to chemotherapy or to the HER2-directed therapy; the combination is synergistic [2]. In patients who progressed on trastuzumab-containing therapy, continued trastuzumab with chemotherapy was superior to chemotherapy alone, suggesting that synergy persists despite prior progression on the same HER2-directed treatment [45,46]. To date, the mechanisms governing a poor response to HER2-directed therapy are not fully understood, but cultured cell lines have offered important insights. A poor response to HER2-directed therapy could be due to three properties of the HER2 receptor: (1) *Receptor accessibility*. Clinical mAbs cannot bind the truncated form of HER2, p95HER2, because it does not contain mAb-binding epitopes on the extracellular region [47,48,49,50]. Additionally, mAbs cannot bind to HER2 that is sterically occluded by large (glyco)proteins [51,52]. (2) *Receptor heterogeneity*. Breast tumors exhibit spatial (cell-to-cell) and temporal heterogeneity, which can affect the response to HER2-directed therapy [53,54,55,56]. In a fraction of cases, the HER2 status changes following neoadjuvant therapy [57]. (3) *Receptor organization at the plasma membrane*. HER2 forms both functional homodimers and heterodimers with a diverse array of receptors [58,59,60,61,62,63,64,65,66,67,68,69,70,71,72]. While homodimers of HER2 produce a weak downstream signal, HER2 heterodimers generate a strong signal [73,74]. 

Poor response to HER2-directed therapy has been linked to the dysregulation of signaling pathways downstream from the HER2 receptor. First are alterations to the PTEN-PI3K/AKT and MEK/ERK [75,76,77,78] pathways; these pathways regulate processes such as apoptosis, metabolism, cell proliferation, and cell growth. The second is a decrease in the levels of the cyclin-dependent kinase inhibitor p27 [79]; p27 regulates cell proliferation, cell motility, and apoptosis. Additionally, an upregulation of receptor tyrosine kinases can, in turn, alter HER2 signaling [62,63,65,71,80,81,82,83,84,85,86]. Recently, these downstream signaling pathways appeared to be impacted by the nano-organization of HER2 along the plasma membrane. Significant clustering of HER2 on the plasma membrane can lead to (biologic agent-induced) inactivation of receptors or the rapid internalization and degradation of receptors [87,88,89,90]. Consistently, upon crosslinking, the endocytosis of HER2 from the plasma membrane is increased in trastuzumab-sensitive SK-BR-3 and BT-474 cells, but not in trastuzumab-resistant BT-474^R^ cells [91].

Although HER2-directed therapy has increased overall patient survival, there still remains a major challenge in treating HER2-positive breast cancers. Clinical HER2 assays cannot accurately identify which HER2-positive breast cancers will respond to therapy [10,11,12,13,14,15,16,17]. Recent research, enabled by emerging techniques, has revealed the importance of the molecular distribution of receptors on the cellular plasma membrane; the receptor nano-organization may govern downstream signaling processes and, ultimately, cellular function. One strategy to define the receptor nano-organization is via a fluorescence-based super-resolution microscopy approach called quantitative single molecule localization microscopy (qSMLM). It is particularly relevant to HER2-positive breast cancer: qSMLM can count plasma membrane-localized receptors, report on the nano-organization of these receptors, and define their nano-scale heterogeneity. Several studies in cultured cell lines and patient tissues have applied various super-resolution microscopy techniques to assess the HER2 protein distribution and copy number [92,93,94,95]. Previously, we observed significant changes in HER2 patterning when cultured cells were treated with chemotherapeutic agents [96]. We also used this approach to quantitatively assess HER2 in patient tissues [97]. Here, we extend this qSMLM approach by assessing HER2 in both therapy-sensitive and -resistant settings.

Using qSMLM, we first probed signatures associated with trastuzumab resistance in four cultured cell lines. Using trastuzumab and pertuzumab as fluorescent probes, we assessed the detected HER2 density and nano-organization. Next, we used touch prep-qSMLM [97] to assess plasma membrane-localized HER2 in core biopsies from six women with breast cancer prior to a neoadjuvant treatment that included chemotherapy, trastuzumab, and pertuzumab. In cultured cell lines and patient samples, both the detected HER2 density and clustering appeared to be important for therapy sensitivity/response.

## 2. Materials and Methods

### 2.1. Coverslip Cleaning

First, 25 mm diameter coverslips (Thermo Fischer Scientific, Cat# NC9560650, Waltham, MA, USA) were cleaned as previously described [98]. Cleaned coverslips were stored in sterile 35 × 10 mm dishes (Thermo Fischer Scientific, Cat# 153066) and wrapped in aluminum foil until used for cell culture or touch prep. 

### 2.2. Cell Culture 

BT-474, BT-474 clone 5 [99], and SK-BR-3 cells (American Type Culture Collection, Manassas, VA, USA) and JIMT-1 (AddexBio, San Diego, CA, USA) were cultured at 37 °C and 5% CO_2_ in phenol-red-free DMEM (Thermo Fischer Scientific, Cat# 3105303) supplemented with 10% fetal bovine serum (VWR, Cat# 97068-091, Radnor, PA, USA), 1 mM sodium pyruvate (Invitrogen, Cat# 11360070, Waltham, MA, USA), and 4 mM GlutaMAX (Thermo Fischer Scientific, Cat# 35050061). For imaging experiments, approximately 250,000 cells were seeded on coverslips previously coated with human fibronectin protein (50 µg/mL final concentration, R&D systems, Cat# 1918-FN-02M, Minneapolis, MN, USA). 

### 2.3. Antibodies and Fluorescent Dye Conjugation

Trastuzumab and pertuzumab (Genentech) were used for staining the extracellular domains of HER2. Anti-cytokeratin 7 (Abcam, Cat# ab181598, Cambridge, UK) and goat anti-rabbit (Abcam, Cat# ab6702) were used to detect epithelial cells. Trastuzumab and pertuzumab were conjugated to Alexa Fluor 647 (AF647, Invitrogen, Cat# A20006), and goat anti-rabbit was conjugated to Alexa Fluor 405 (AF405, Invitrogen Cat# A30000) using previously published methods [97]. According to NanoDrop measurements, in all cases approximately one dye per antibody (degree of labeling ~1) was obtained for trastuzumab and pertuzumab.

### 2.4. Photophysical Characterization of Fluorescent Probes

To determine the photophysical properties of the fluorescent probes (maximum dark time and the average number of localizations per fluorescent probe), we used a surface assay for molecular isolation (SAMI) [96]. Surfaces with sparse trastuzumab- or pertuzumab-AF647 were imaged immediately in dSTORM imaging buffer [100].

### 2.5. Flow Cytometry in Cultured Cell Lines

Cells were grown to ~90% confluency and harvested by trypsinization. After washing, a total of 3 × 10^5^ cells were incubated by rotation for 30 min at 4 °C with trastuzumab-AF647 using a fixed concentration of 10 µg/mL. The same procedure was also applied for untreated cells to evaluate the background fluorescence signal; AF647-conjugated goat anti-human IgG was used as a negative control. Cells were then pelleted by centrifugation at 264× *g* for 5 min, washed three times, and resuspended in 200 µL of buffer (PBS 2% BSA). Experiments were run on an LSRII FortessaX20 Analytical Cytometer where a minimum of 10,000 events were recorded for each sample in triplicate. The data were then analyzed using FlowJo ^TM^ v10 Software.

### 2.6. Immunofluorescence Staining of Cultured Cells

Following a two-day seed on coverslips, cells were fixed with 4% (*w*/*v*) paraformaldehyde and 0.2% (*w*/*v*) glutaraldehyde (EMS, Cat# 157-8 and 16019, respectively, Hatfield, MA, USA) in PBS for 30 min at room temperature. Fixation was then quenched with 25 mM glycine in PBS for 10 min followed by three washes with blocking buffer (BB, 5% bovine serum albumin (RPI, Cat# A30075-100.0, Mount Prospect, IL, USA) and 0.02% Tween-20 (G-Biosciences, Cat# DG011, St. Louis, MO, USA) in PBS for 20 min at room temperature. Cells were then incubated with 50 nM trastuzumab-AF647 or pertuzumab-AF647, diluted in BB, for 1 h at room temperature and covered from light. Following staining, cells were washed twice with BB, four times with PBS, and post-fixed with 4% (*w*/*v*) paraformaldehyde and 0.2% (*w*/*v*) glutaraldehyde in PBS for 10 min at room temperature. Fixation was quenched using 25 mM glycine in PBS for 10 min at room temperature. Cells were washed three times with PBS and incubated with 0.1 µm TetraSpeck microspheres (Invitrogen, Cat# T7279) in PBS for 5 min. Coverslips were placed in Attofluor cell chambers and imaged immediately in dSTORM imaging buffer [100]. 

. 

### 2.7. Human Subjects and Core Biopsy Collection

Tissue samples were collected under City of Hope (COH) Institutional Review Board (IRB) number 16079. Dedicated core needle biopsies were performed by a radiologist under ultrasound guidance prior to the initiation of neoadjuvant therapy. Immediately following collection, the tissue was transferred to a pathologist (DS) for cell collection, which was performed in the radiology suite. Portions of the tissue that were grossly consistent with carcinoma were identified and were separated from, e.g., blood clot. The tissue was lightly touched to sterile gauze to dry it, and touch prep was accomplished by gently holding the biopsy material with a forceps and dabbing the tissue against the coverslip multiple times. Multiple portions of tissue (at least two) were sampled for each biopsy specimen.

### 2.8. Tissue Touch Prep and Immunofluorescence Staining

Cleaned coverslips were incubated with 100 µL of poly-L-lysine solution (Millipore Sigma, Cat# P4707) for 15 min, washed three times with sterile DI water, and dried at room temperature for 10 min. Core biopsy tissue was dabbed several times on coverslips to leave behind a layer of cells. The monolayer left behind on the coverslip was allowed to incubate on the coverslips for 5 min to facilitate adhesion before the addition of any liquids. The sample was then fixed with 4% (*w*/*v*) paraformaldehyde and 0.2% (*w*/*v*) glutaraldehyde in PBS for 30 min at room temperature. Fixation was quenched with 25 mM glycine in PBS for 10 min, followed by three washes with BB (however, tissue BB was supplemented with 0.1% Tween-20 instead of the 0.02% used with the cultured cell lines). The tissue was then incubated with BB for 20 min at room temperature. After blocking, the tissue samples were incubated with 50 nM trastuzumab-AF647 and 2 µg/mL anti-cytokeratin 7 Ab, diluted in BB, for 1 h at room temperature, covered from light. Following primary antibody incubation, the tissue was washed with BB and incubated with 2 µg/mL goat anti-rabbit-AF405 for 45 min at room temperature. The tissue was washed twice with BB and four times with PBS. Then, it was postfixed with 4% (*w*/*v*) paraformaldehyde and 0.2% (*w*/*v*) glutaraldehyde in PBS for 10 min, followed by quenching with 25 mM glycine in PBS for 10 min at room temperature. The samples were imaged immediately using dSTORM imaging buffer [100]. 

### 2.9. dSTORM Setup and Image Acquisition

For patient biopsy samples, imaging was performed on a 3D N-STORM super-resolution microscope (Nikon, Minato City, Tokyo, Japan), consisting of a Ti-E inverted microscope with an E-TIRF illuminator and a Piezo Z stage on a vibration isolation table. This N-STORM system was equipped with a CFI 100XH oil objective Apo, TIRF, NA 1.49, N-STORM and λ/4 lenses, quad cube C-NSTORM (Chroma, Cat# 97355, Bellows Falls, VT, USA), a Perfect Focus System, an MLC-MBP-ND laser launch (Agilent, Santa Clara, CA, USA), and an iXON Ultra DU-897U EMCCD camera (Andor Technology, Belfast, UK). For cultured cell lines, we used an upgraded N-STORM system with a Nikon Ti2-E inverted microscope, Nikon N-STORM TIRF illuminator with a 2× magnification lens, and Nikon LU-NF laser launch. The laser powers used from the Ti-E and Ti2-E systems to excite the AF647 dyes were 140 mW and 89 mW, respectively, as measured out of the optical fiber. The laser power for the activation of AF405 dye was 5–10 mW, as measured out of the optical fiber. We acquired 60,000 frames for the cultured cell lines and 20,000 frames for the patient biopsy samples using an exposure of 10 ms. The data were acquired using NIS Elements software (Nikon, Version 4.3 for Ti-E and 5.21.01 for Ti2-E).

### 2.10. Image Processing and Analysis

Following image acquisition, the data were first processed using NIS Elements to localize each molecule in each frame. The patient biopsy data were processed with a minimum height of the fluorophore point spread function of 5000 and a minimum photons of 700. For the cultured cell line data, these thresholds were set to 5000 and 1400, respectively. The identification settings for both datasets were as follows: 200 nm minimum peak width, 400 nm maximum peak width, 300 nm initial fit width, 1.3 maximum axial ratio, and 1-pixel maximum displacement. Additionally, the photons per count was set to 0.016 and 0.018 for the patient biopsy samples and cultured cell lines, respectively, based on the camera settings. 

The localization data were exported from the NIS Elements software into MATLAB (MathWorks, Version 2021a) for further processing. Fiducial beads were used to correct for the drift that occurred during imaging. For plotting the neighbors map for the cultured cell line data, a density filter was applied to remove the beads using the threshold of 100 counts in a radius of 60 nm. For the patient biopsy data, a density filter was applied to remove artificial clusters [101] with more than 50 counts in a radius of 70 nm. For both datasets, the lateral localization precision (σ) was calculated by NIS Elements based on the photon distribution, and localizations with a precision larger than the 98th percentile were discarded. The localization density and clustering (auto-correlation) functions were computed using custom MATLAB code. Three to four square ROIs of 16 µm^2^ (cultured cell lines) or one to five square ROIs of 20 µm^2^ (patient biopsy data) were placed across cell areas to detect the total number of localizations. The number of localizations was divided by the average number of localizations per molecule, α, to obtain the detected number of molecules within each ROI. 

Within these ROIs, fast Fourier transform was used to compute the auto-correlation functions (using PC analysis); for clustered ROIs, the proteins per cluster and cluster radii were subsequently calculated [102,103]. For the same clustered ROIs, the fraction of clustered receptors and the number of clusters were quantified using a k-means-like clustering algorithm [96,97]. This algorithm uses the average σ, the cluster radius obtained from the PC analysis, and the maximum fluorophore dark time to determine the fraction of receptors in clusters. If molecules met spatiotemporal requirements, they were counted as part of a cluster. For this clustering analysis, we report a cluster as containing more than two HER2 receptors (to account for isolated HER2 homodimers). A neighbor counts map was determined by taking an individual localization and locating neighboring localizations within a 70 nm radius.

### 2.11. Statistical Analysis

A minimum of 14 cells were imaged for the cultured cell line data, and a minimum of 12 cells were imaged for the patient biopsy data. For the cultured cell line data, imaging was conducted in triplicate (on three separate days). For the patient biopsy samples, we imaged at least two separate coverslips for each individual patient. All data in graphs are summarized by reporting mean values and SEM. *p* values were calculated for cultured cell lines and patient biopsy samples in Excel using a one-tailed Student’s t-test with a heteroscedastic two-sample unequal variance type. Additionally, to evaluate sampling, we randomly split all ROIs for a given output (density, cluster radius, HER2/cluster, the fraction of clustered HER2, or the number of HER2 clusters/ROI) into two groups and calculated the *p* values between the two groups (*p*-value_split_). This was carried out using a randomization function followed by splitting and calculation by a two-tailed Student’s t-test with an unequal variance type in MATLAB. We ran this randomization and t-test five times and took the average for reporting. In all cases, no significant difference was observed between the two groups (*p*-value_split_ ≥ 0.1). For all data, we also calculated median, the coefficients of variation (the relative dispersion of detected densities around the mean value), kurtosis (the measure of tailedness of the distribution that can identify if tails have extreme values), and skewness (the measure of symmetry/asymmetry in the dataset) using Graph Pad Prism 8. All data for cultured cell lines are reported in Appendix A, while the values for patient biopsies are provided in Appendix A. 

## 3. Results

### 3.1. Molecular Imaging of HER2 in Cultured Breast Cancer Cell Lines

We assessed HER2 in cultured cell lines that express HER2 protein (Appendix A). Using single molecule imaging, we compared the molecular features of plasma membrane-localized HER2 in two model systems. The first comprised the HER2-positive luminal B molecular subtype [104], which is estrogen receptor (ER)- and/or progesterone receptor (PR)-positive. The second comprised the HER2-enriched molecular subtype [104], which is ER-negative and PR-negative. Our luminal B model system encompassed the trastuzumab-sensitive BT-474 and trastuzumab-resistant BT-474 clone 5 (BT-474^R^) cell lines. We purchased BT-474^R^ from ATCC; the cells were developed by culturing in the presence of trastuzumab until the emergence of resistant clones. This cell line exhibits enhanced phosphorylation of AKT and reduced nuclear expression of the cyclin-dependent kinase inhibitor p27 upon treatment with trastuzumab [99]. Our second model system encompassed cell lines with a phenotype consistent with HER2-enriched cells: trastuzumab-sensitive SK-BR-3 and trastuzumab-resistant JIMT-1 cells. Historically, JIMT-1 cells were isolated from a patient who showed resistance to trastuzumab therapy. In this cell line, trastuzumab cannot efficiently bind to HER2; the epitope is masked by a bulky cell surface glycoprotein mucin-4 [51] and/or the CD44 ligand hyaluronan [105]. 

In these four cell lines, we detected HER2 using two clinical antibodies, trastuzumab and pertuzumab. Each binds to different domains of extracellular HER2 (Appendix A). Trastuzumab binds to domain IV and may inhibit ligand-independent HER2 homodimerization. Pertuzumab binds to domain II and can inhibit ligand-induced HER2 heterodimerization [106,107]. Performing independent studies with each antibody allowed us to assess two distinct HER2 pools. Using an optimized protocol [97], we fluorescently labeled the antibody with Alexa Fluor 647 (AF647) dye. In all our experiments, the degree of labeling was approximately 1 (Materials and Methods). We then used an established protocol (surface assay for molecular isolation, SAMI) [96] to assess the photophysical properties of the fluorescently labeled probes. For our imaging conditions, the average number of localizations per single fluorescently labeled antibody (trastuzumab or pertuzumab) was 5, and the maximum dark time was 300 s (Appendix A).

Prior to imaging, cultured cells were fixed using a protocol that does not induce artifacts but properly immobilizes proteins [102,108]. After the immunofluorescence staining of cells, SMLM was employed to assess HER2. Representative SMLM images (Figure 1) show clear differences in the detected HER2 densities between the various cultured cell lines and fluorescently labeled antibodies. The images were plotted using a neighbor counts map (Materials and Methods) to show the relative clustering of localizations. Purple color indicates no clustering of localizations, blue color indicates low clustering of localizations, and yellow color indicates high clustering of localizations. When trastuzumab was used as a probe (representative images in Figure 1a,b, top), we visualized increased HER2 densities and clustering in the trastuzumab-sensitive cells (BT-474 and SK-BR-3) compared to their respective trastuzumab-resistant cells (BT-474^R^ and JIMT-1). When pertuzumab was used as a probe (representative images in Figure 1a,b, bottom), the HER2 densities and clustering between BT-474, BT474^R^, and SK-BR-3 cells visually did not show pronounced differences; however, JIMT-1 cells showed markedly reduced HER2 density. Appendix A includes the localization precision for this dataset.

### 3.2. Quantitative SMLM in luminal B Cultured Breast Cancer Cell Lines

We next analyzed SMLM imaging data in our luminal B model to quantify both the detected density of HER2 and its nano-organization. The detected density was determined by dividing the total number of localizations by the average number of localizations per fluorescent probe [96,97]. The HER2 cluster radius and the number of HER2 molecules per cluster were determined for regions of interest (ROIs) with non-random (clustered) HER2 organization using the value for the average number of localizations per fluorescent probe (from SAMI) and pair-correlation (PC) analysis [97,102,103]. For the same ROIs, we determined the number of HER2 clusters with more than two molecules and the fraction of HER2 molecules in those clusters relative to all detected HER2 molecules. To calculate these parameters, we used values for maximum dark time (from SAMI), values for the average number of localizations per fluorescent probe (from SAMI), values for cluster size (from PC analysis), and a k-means-like clustering algorithm [96,97]. This approach has been previously validated with a combination of cell line data and simulations [97].

Quantitative information on HER2 density and distribution at the plasma membrane was obtained from three independent measurements with sufficient cell sampling (in four cell lines, the *p-*value_split_ [97] for all measured variables with both probes was above 0.1). When trastuzumab was used as a probe, quantitative analysis of SMLM data revealed substantial differences between trastuzumab-sensitive BT-474 cells and trastuzumab-resistant BT-474^R^ cells. The average detected HER2 density in BT-474 cells was significantly higher compared to BT-474^R^ cells (221 vs. 182 molecules/µm^2^, Figure 2a and Appendix A); BT-474 cells also had a higher coefficient of variation for the detected densities (43 vs. 28%). Additionally, the two cell lines exhibited differences in HER2 nano-organization. While the cluster radii (17.1 and 16. 3 nm) and average number of HER2 molecules in a cluster (2.7 and 2.8) were similar between the two cell lines (Figure 2b,c and Appendix A), both the number of HER2 clusters per ROI and the fraction of clustered HER2 molecules were significantly higher in BT-474 cells compared to BT-474^R^ cells (Figure 2d,e and Appendix A). On average, BT-474 cells had 1264 clusters per ROI and 48% of HER2 molecules were in these clusters; BT-474^R^ cells had 928 clusters per ROI and 44% of HER2 molecules were in these clusters. Interestingly, when pertuzumab was used as a probe, we did not observe significant differences between the two cell lines in terms of either detected HER2 densities or nano-organization (Figure 2f–j and Appendix A). Accordingly, we observed substantial differences between trastuzumab and pertuzumab detection. In both cell lines, detection with pertuzumab resulted in a significantly lower detected HER2 density, a smaller HER2 cluster radius, a lower fraction of clustered HER2, and a smaller number of detected HER2 clusters (Figure 2 and Appendix A). 

### 3.3. Quantitative SMLM in HER2-Enriched Breast Cancer Cell Lines

We then performed similar analyses in the HER2-enriched model system, where we compared trastuzumab-sensitive SK-BR-3 cells to trastuzumab-resistant JIMT-1 cells. When trastuzumab was used as a probe, we observed significant differences between the two cell lines in terms of the detected HER2 densities and nano-organization. Figure 3a and Appendix A illustrate a substantial difference in the average detected HER2 densities (181 vs. 22 molecules/µm^2^); the coefficient of variation was higher in sensitive cells. Compared to JIMT-1 cells, SK-BR-3 cells, on average, had a significantly higher cluster radius (17.4 vs. 14.9 nm), higher number of HER2 molecules in clusters (2.9 vs. 2.0), higher fraction of clustered HER2 (52 vs. 36%), and higher number of detected HER2 clusters per ROI (1222 vs. 117) (Figure 3b–e and Appendix A). When pertuzumab was used as a probe, SK-BR-3 cells also exhibited higher average detected HER2 densities (57 vs. 10 molecules/µm^2^) and a higher number of HER2 clusters per ROI (333 vs. 62) (Figure 3f,j and Appendix A). Detection with the pertuzumab probe yielded similar values for the cluster radius, number of HER2 molecules in clusters, and fraction of clustered HER2 (Figure 3g–i and Appendix A). According to these cumulative results, employing qSMLM with trastuzumab as the probe may provide signatures associated with trastuzumab sensitivity in cultured cell lines, namely, high HER2 density and clustering. 

### 3.4. Detected HER2 Density and Nano-Organization in Patient Tissue Specimens

We hypothesized that the HER2 molecular features observed in cultured cell lines may also be present within patient specimens, thereby representing a potential signature of therapy response. We previously developed a touch prep-qSMLM [97,109] approach to rigorously assess the detected HER2 density and nano-organization in large areas of intact membranes from patient excisions. Importantly, the detected HER2 densities from qSMLM had a significant positive correlation with the HER2 copy numbers from the clinical FISH assay [97]. Here, we extended the approach to core biopsy samples obtained prior to the initiation of neoadjuvant therapy with docetaxel, carboplatin, trastuzumab, and pertuzumab (TCHP). This is the point in the treatment timeline where information regarding the response to therapy could inform subsequent therapeutic decisions. We assessed tumors from six women with breast cancer; in one woman two tumor sites were sampled (denoted here as S1 and S2). At the time of biopsy, five breast cancers had IHC (3+); one breast cancer had negative FISH and IHC (2+). The patient characteristics are shown in Appendix A. 

The human breast cancer biopsy specimens used for qSMLM were collected as part of a COH IRB-approved trial (IRB # 16079) examining the use of PET imaging to predict therapy response. Per the protocol, women were given one dose of radiolabeled trastuzumab prior to PET imaging. Several days later (range = 1–14 days), the women underwent ultrasound-guided core biopsy under the care of a breast radiologist. Biopsy specimens were immediately transferred to a pathologist for cell collection to minimize the ischemic time. The portions of the biopsy material that were grossly consistent with carcinoma were identified and lightly touched to sterile gauze to dry the tissue. We then used a touch prep technique [110] to adhere monolayers of cancer cells from core biopsies to poly-L-lysine-coated coverslips. This was accomplished by gently holding the biopsy material with forceps and dabbing the tissue against the coverslip multiple times. At least two regions were taken from each biopsy sample. The samples were then immediately fixed and incubated with trastuzumab-AF647 and an antibody against the epithelial marker, cytokeratin 7, labeled with Alexa Fluor 405. After analytical sample processing, the samples were immediately imaged. The cell morphology was assessed in brightfield, and epithelial cells were distinguished from stromal cells [111] using the 405 nm signal. Trastuzumab-AF647 was then used in qSMLM to detect trastuzumab-accessible extracellular HER2 in epithelial cells. For each biopsy specimen, we calculated the detected HER2 density and nano-organization. A minimum of 12 cells were imaged, and sufficient sampling was confirmed (*p*-value_split_ was above 0.1 for all variables, Appendix A). At the time of surgery (approximately 6 months following core biopsy collection and after all qSMLM data were computed), clinical outcomes and residual cancer burden (RCB) indices were noted. The RCB index (RCB-0 to RCB-III) was designed to quantify residual disease in the tumor bed and lymph nodes after neoadjuvant therapy; it is a well validated prognostic indicator for breast cancer [112,113]. Our sample cohort comprised the following: P18 had RCB-I, P20 had RCB-II, and all other women had achieved a pathological complete response (pCR) equivalent to RCB-0. Of note, P26 had residual ductal carcinoma in situ (DCIS). 

A SMLM image for a biopsied HER2-positive cancer cell is shown in Figure 4a. Representative SMLM images for each individual woman are shown in Appendix A. Although most cells had relatively uniform distributions of HER2, a few (e.g., P21 and P26, Appendix A) had noticeable non-uniform densities. Interestingly, HER2-enriched cell protrusions were also observed in some cells (Figure 4a, indicated by arrows). This is consistent with previous reports of HER2 enrichment in membrane protrusions [114]. The average detected HER2 densities in these women ranged from 49 to 157 molecules/µm^2^. The lowest values were observed in patients P18 (RCB-I), P20 (RCB-II), and P26 (pCR with DCIS): 49, 58, and 59 molecules/µm^2^, respectively (Figure 4b and Appendix A). Interestingly, these three women also had the largest coefficients of variations for detected HER2 densities. The two regions taken from the P25 biopsy sample had significantly different densities (99 and 137 molecules/µm^2^), indicating that HER2 expression was different at the two tumor sites. There was no association between the heterogeneity in detected densities and the clinical response (Appendix A). 

We next determined the nano-organization parameters. Of note, in P20 (negative FISH and IHC 2+) we detected only 4 regions with HER2 clusters; the remaining 54 regions had a random HER2 distribution. The average cluster radii (Figure 4c and Appendix A) varied between 18 nm and 38 nm. The number of molecules per cluster (Figure 4d) was slightly lower in women that did not achieve pCR: P18 (RCB-I) and P20 (RCB-II) had on average 2.3 and 1.8 molecules per cluster, respectively, whereas the other women had average values ranging between 2.6 and 3.8 molecules per cluster. The fraction of clustered HER2 molecules was significantly reduced in patients P18 and P20 (33 and 31%, respectively) compared to the other women with pCR (ranging from 54 to 73%, Figure 4e). The number of HER2 clusters per ROI showed even more pronounced differences between women who did and did not achieve pCR (Figure 4f). On average, P18 (RCB-I) had 199 clusters per ROI and P20 (RCB-II) had 118 clusters per ROI, while the women who achieved pCR had between 638 and 909 clusters per ROI. Therefore, both the detected HER2 densities and clustering values may be important when considering the response to therapy. 

## 4. Discussion

The clinical antibodies trastuzumab and pertuzumab bind to extracellular domains of HER2 (Appendix A). Using either trastuzumab-AF647 or pertuzumab-AF647 as probes for qSMLM, we assessed HER2 in the steady state of four cultured cell lines. We probed HER2-positive luminal B cells, BT-474 (trastuzumab-sensitive) and BT-474^R^ (trastuzumab-resistant), and HER2-enriched cells, SK-BR-3 (trastuzumab-sensitive) and JIMT-1 (trastuzumab-resistant). All cell lines were cultured without antibiotics. According to the obtained images (Figure 1), robust HER2 staining was achieved. Consistent with flow cytometry [115] and our previous report [97], BT-474 cells had a higher detected HER2 density compared to SK-BR-3 cells (Figure 2 and Figure 3). 

Trastuzumab binds to domain IV of HER2, close to the plasma membrane. In contrast, pertuzumab binds to domain II, which is important for interactions with two other members of the human epidermal growth factor receptor family, HER1 and HER3 [106]. We used qSMLM to quantify how the binding to specific HER2 epitopes impacted the detected HER2 densities; we also compared the effects in trastuzumab-sensitive vs. -resistant cell lines. To these ends, we calculated the ratio of detected HER2 obtained with the two probes (trastuzumab/pertuzumab). In the four cell lines, the ratio was greater than one (trastuzumab detected more HER2 molecules compared to pertuzumab). The ratios for the two trastuzumab-sensitive cell lines were 3.1 (SK-BR-3) and 2.6 (BT-474). The ratios for the trastuzumab-resistant cell lines were 2.2 (JIMT-1) and 2.0 (BT-474^R^). The higher ratios in sensitive cells vs. resistant cells could reflect the number of HER2 molecules engaged in heterodimers. According to recent molecular dynamic simulations, trastuzumab can bind to HER2 homodimers and heterodimers [116]; it does not directly block the dimerization interface. Conversely, pertuzumab blocks the heterodimerization interface [106]. While we cannot rule out differences related to the analytical protocol (e.g., sensitivity to fixation), steric effects may be in play for the two probes that bind to distinct HER2 epitopes. The two probes likely detect distinct HER2 pools.

Using qSMLM, with molecular sensitivity we assessed how HER2 density (detected with two clinical Abs) compared between BT-474 and BT-474 ^R^ cells. Trastuzumab, but not pertuzumab (Figure 1 and Figure 2), detected significantly higher HER2 densities for BT-474 cells. Additionally, published data using flow cytometry (using an Ab that did not compete with trastuzumab binding to HER2) suggest that the expression of HER2 is comparable in BT-474^R^ and BT-474 cells [99]. Thus, the differences in trastuzumab-detected HER2 densities with qSMLM may be caused by its reduced efficiency to bind HER2 in BT-474^R^ cells, i.e., steric occlusion of the binding epitope. Both trastuzumab and pertuzumab (Figure 1 and Figure 3) detected significantly higher HER2 densities for SK-BR-3 vs. JIMT-1 cells. While these two cell lines have comparable DNA copy numbers [117,118], SK-BR-3 cells showed detected HER2 densities that were approximately eight-fold higher for trastuzumab and approximately six-fold higher for pertuzumab. One potential explanation for the discordance between the copy numbers and detected densities is inaccessible extracellular HER2 in resistant cell lines. For example, in JIMT-1 cells, HER2 epitopes have been shown to be masked by glycosylated membrane proteins [51,119]. 

When trastuzumab was used as a probe, we also detected significant differences in HER2 clustering. We considered HER2 clusters to have more than two molecules (clustering beyond homodimers). In trastuzumab-sensitive vs. trastuzumab-resistant cells (BT-474 vs. BT-474^R^ and SK-BR-3 vs. JIMT-1), we observed more HER2 clusters and a higher fraction of clustered HER2. Additionally, in SK-BR-3 vs. JIMT-1 cells, we observed a larger average cluster radius and more HER2 molecules per cluster. The differences in nano-organization were significantly attenuated with the pertuzumab probe. While qSMLM (with pertuzumab-AF647) did not detect any significant difference in HER2 clustering parameters for BT-474 vs. BT-474^R^ cells, it detected a significantly higher number of HER2 clusters per ROI in SK-BR-3 vs. JIMT-1 cells.

We have previously shown that touch prep-qSMLM is a powerful tool for assessing HER2 molecules in human tissue specimens [97]. Here, we extended this methodology to core biopsy samples obtained from six women prior to neoadjuvant therapy (chemotherapy combined with trastuzumab and pertuzumab). We detected highly resolved molecular details (e.g., membrane protrusions, Figure 4a) and determined both the detected HER2 densities and clustering characteristics (Figure 4b–f). Due to the number of available patient specimens, our study was limited to a small sample size. However, we found high detected HER2 densities and significant HER2 clustering in women who achieved pCR. Although pCR is not an established surrogate marker for long-term clinical outcomes, it is well-established that women who achieve pCR have a better prognosis than women who do not achieve pCR [120,121]. Although subjects received chemotherapy in addition to trastuzumab and pertuzumab (and thus it is possible that pCR could have been achieved without HER2-directed therapy), our data from human breast cancer biopsies are consistent with data from cultured cell lines. Given that increased clustering of receptors can result in the rapid internalization/degradation of HER2 or the formation of inactive plasma membrane complexes with biologic agents [89,90], the clustering of HER2 may have important therapeutic implications. 

## 5. Conclusions

Our qSMLM data indicate that a pool of plasma membrane-localized HER2 directly detected by trastuzumab (but not pertuzumab) may provide important insight into the therapy response in breast cancer cells. While other mechanisms of resistance may be in play, we have focused on the molecular assessment of HER2 that is accessible by clinical mAbs. Detected HER2 densities could complement the current clinical diagnostic tools by robustly quantifying HER2 protein levels that are available to therapeutic antibodies. Additionally, our data suggest that measurements of detected HER2 density alone cannot predict which women will benefit the most from therapy. While P26 (pCR), P18 (RCB-I), and P20 (RCB-II) had low detected HER2 densities, HER2 clustering was reduced only for P18 and P20. Thus, beyond HER2 expression, molecular receptor organization appears to be a relevant factor. Since such information cannot be obtained using conventional diffraction-limited microscopy, this qSMLM approach may be helpful in complementing other clinical assays to predict which women will respond to therapy.

## Figures and Tables

**Figure 1 cancers-14-02795-f001:**
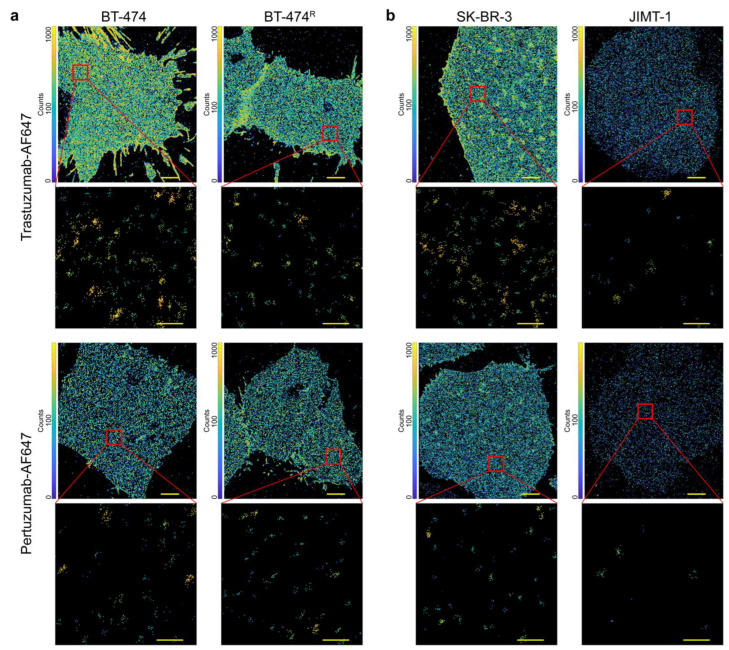
SMLM images of HER2 detected in cultured breast cancer cell lines. (**a**)**.** HER2 was detected using trastuzumab-AF647 (top) or pertuzumab-AF647 (bottom) in luminal B breast cancer cells BT-474 (trastuzumab-sensitive) and BT-474^R^ (trastuzumab-resistant). (**b**)**.** HER2 was detected using trastuzumab-AF647 (top) or pertuzumab-AF647 (bottom) in HER2-overexpressing breast cancer cells SK-BR-3 (trastuzumab-sensitive) and JIMT-1 (trastuzumab-resistant). The localizations are colored based on the number of neighboring localizations within 70 nm (purple is least clustered and yellow is most clustered). Scale bars are 5 µm for full cells and 200 nm for insets. While the figure shows representative images, measurements were repeated three independent times, and 14–15 cells were imaged in each case.

**Figure 2 cancers-14-02795-f002:**
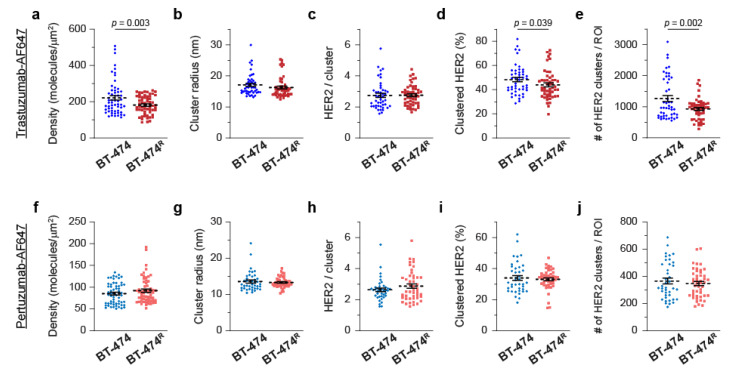
Detected HER2 densities and clustering parameters in cultured luminal B breast cancer cell lines. Average detected HER2 density (**a**) and clustering parameters (**b**–**e**) obtained using detection with trastuzumab-AF647 in BT-474 (blue) and BT-474^R^ (red) cells. When trastuzumab was used as a probe, trastuzumab-sensitive cells (compared to trastuzumab-resistant cells) on average had a significantly higher detected density, fraction of HER2 molecules in clustered regions, and number of HER2 clusters per ROI. In total, 58 ROIs for BT-474 and 60 ROIs for BT-474^R^ were assessed. Clustering parameters were calculated based on 47 clustered ROIs for BT-474 and 51 clustered ROIs for BT-474^R^ cells. Average detected HER2 densities (**f**) and clustering parameters (**g**–**j**) obtained using detection with pertuzumab-AF647 in BT-474 (blue) and BT-474^R^ (red) cells. When pertuzumab was used as a probe, trastuzumab-sensitive and trastuzumab-resistant cells had comparable detected densities and clustering parameters. In total, 60 ROIs for BT-474 and 60 ROIs for BT-474^R^ were assessed. Clustering parameters were calculated based on 40 clustered ROIs for BT-474 and 47 clustered ROIs for BT-474^R^ cells. Dashed lines represent the mean, and all error bars represent SEM. Numerical details and statistics are provided in Appendix A.

**Figure 3 cancers-14-02795-f003:**
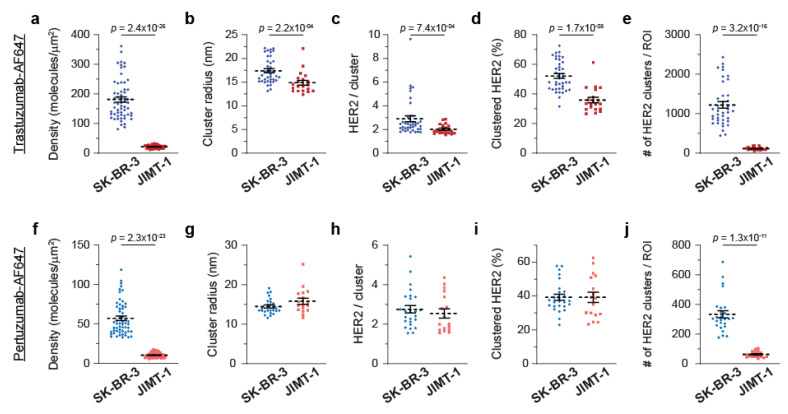
Detected HER2 densities and clustering parameters in cultured HER2 overexpressing breast cancer cell lines**.** Average detected HER2 density (**a**) and clustering parameters (**b**–**e**) obtained using detection with trastuzumab-AF647 in SK-BR-3 (blue) and JIMT-1 (red) cells. When trastuzumab was used as a probe, trastuzumab-sensitive cells (compared to trastuzumab-resistant cells) had a significantly higher detected density and an increase in HER2 clustering according to all clustering parameters. In total, 60 ROIs for SK-BR-3 and 59 ROIs for JIMT-1 were assessed. Clustering parameters were calculated based on 40 clustered ROIs for SK-BR-3 and 20 clustered ROIs for JIMT-1 cells. Average detected HER2 densities (**f**) and clustering parameters (**g**–**j**) obtained using detection with pertuzumab-AF647 in SK-BR-3 (blue) and JIMT-1 (red) cells. When pertuzumab was used as a probe, trastuzumab-sensitive cells (compared to trastuzumab-resistant cells) had significantly higher average detected density and number of HER2 clusters per ROI. In total, 55 ROIs for SK-BR-3 and 60 ROIs for JIMT-1 were assessed. Clustering parameters were calculated based on 26 clustered ROIs for SK-BR-3 and 17 clustered ROIs for JIMT-1 cells. Dashed lines represent the mean, and all error bars represent SEM. Numerical details and statistics are provided in Appendix A.

**Figure 4 cancers-14-02795-f004:**
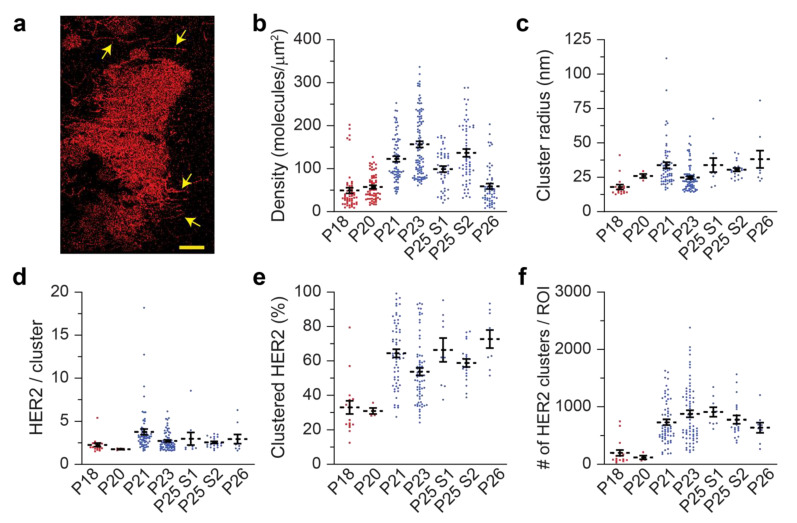
Detected HER2 density and clustering parameters in human breast cancer core biopsies. (**a**). SMLM image of a HER2-positive cell from a core biopsy sample with HER2-enriched cell protrusions marked with arrows. Scale bar is 5 µm. (**b**) Average detected HER2 densities in patient specimens obtained using trastuzumab-AF647. P21, P23, and P25 (pCR, T0 N0 post-op) had significantly higher densities compared to P18 (RCB-I, T1aN0 post-op), P20 (RCB-II, T1b N1a post-op), and P26 (pCR, Tis N0 post-op). (**c**–**f**). Average HER2 clustering parameters in patient specimens obtained using trastuzumab-AF647. P21, P23, P25, P26 had significantly higher fractions of HER2 molecules in clustered regions and numbers of HER2 clusters per ROI compared to P18 and P20. Clustered ROIs were used for analysis: 17 ROIs for P18, 4 ROIs for P20, 61 ROIs for P21, 70 ROIs for P23, 9 ROIs for P25 S1, 21 ROIs for P25 S2, and 9 ROIs for P26. Red points indicate women with RCB and blue points indicate women with pCR. Dashed lines represent the mean, and all error bars represent SEM. Numerical details and statistics are provided in Appendix A.

## Data Availability

The data presented in this study is available in this article (and Appendix A).

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
