# Peer review of "Molecular Assessment of HER2 to Identify Signatures Associated with Therapy Response in HER2-Positive Breast Cancer"

_cancers, 2022, doi:10.3390/cancers14112795_

Round 1

Reviewer 1 Report

Maddox et al reported that nano-organization of HER2 in trastuzumab sensitive and resistant cell lines of Luminal B and HER2+ breast cancer subtypes using a quantitative single molecule localization microscopy (qSMLM). Higher HER2 clustering and densities were found in sensitive cells compared to resistant cells. Finally, HER2 density and nano-organization was also studied in patient tissue specimens. Overall this study designed for the use of advanced qSMLM to overcome the current clinical diagnostic assays for better identification of HER2 response targeted therapies in breast cancer patients. However, a few potential major points are not addressed in this manuscript.

  1. Authors would show the expression of HER2 at protein levels in trastuzumab resistant and sensitive cells.
  2. It is not clear that the decreased HER2 densities and clustering in resistant cells is due to low HER2 expression or low binding efficiency of trastuzumab to HER2 which is masked by cell surface glycoproteins. Whereas, increased HER2 densities in sensitive cells is because of high HER2 expression.
  3. The potential drawback of this manuscript is conducting experiments in one luminal B and HER2+ cell lines. It is better to show more than two cell lines.
  4. Data has not been showed to prove the characteristic features of resistant or sensitive cells of HER2+ and luminal B.

Reviewer 2 Report

Reviewer comments:

Comments to the Author

The authors have investigated quantitative single molecule localization microscopy to assess molecular features of HER2 in a therapy-responsive setting. Using fluorescently labeled trastuzumab as a probe, authors have compared the molecular features of HER2 in trastuzumab sensitive (BT-474, SK-BR-3) and trastuzumab-resistant (BT-474R, JIMT-1) cultured cell lines.

The study is impressive, and manuscript is for the most part well written, the experimental progression was logical, and the data provided was comprehensive, well validated and presented clearly.

I have the following specific minor concerns.

  • Scale bars in some of the images Figure 1 are missing. Please include them.
  • Please undergo a thorough check of the manuscript for typographical and grammatical errors.

Reviewer 3 Report

Reviewer comments:

Comments to the Author

This manuscript describes “Molecular Assessment of HER2 to Identify Signatures Associated with Therapy Response in HER2-Positive Breast Cancer” by Adam et., al. The study is well planned and logically presented.

This work is of interest to scientists studying how molecular features of HER2 can be implied in breast cancer therapies. 

As such the data presented herein are exemplar in demonstrating: i) the substantial quantitative single molecule localization microscopy to assess molecular features of HER2 in a therapy-responsive setting. ii) Using fluorescently labeled trastuzumab as a probe to compare the molecular features of HER2 in trastuzumab sensitive (BT-474, SK-BR-3) and trastuzumab-resistant (BT-474R, JIMT-1) cultured cell lines.

All in all I found the experimental model used very helpful in demonstrating this important link between: HER2 clustering -> potential signature of therapy-responsive disease.

Below are few minor concerns aiming to improve the work.

  • In figure 1, information about number of samples used to conduct this experiment is not provided. Authors are advised to provide N values for each experiment in the figure legends. How molecular imaging data is reproducible?
  • Please undergo a thorough check of the manuscript for typographical and grammatical errors.

Round 2

Reviewer 1 Report

I recommend this manuscript to accept in the present form.